# Mexican and Spanish university students' Internet addiction and academic procrastination: Correlation and potential factors

Inmaculada Aznar-Díaz[1☺], José-María Romero-Rodríguez[1☺]*, Abel García-González[2☺], María-Soledad Ramírez-Montoya[2☺]

1 Department of Didactics and School Organization, University of Granada, Granada, Spain, 2 Humanities and Education School, Tecnologico de Monterrey, Nuevo Leon, Mexico

☺ These authors contributed equally to this work.
* romejo@ugr.es

**Data Availability Statement:** The data matrix has been uploaded as a supplementary file.

## Abstract

The 21st-century problem of Internet addiction is increasing globally, but especially among university students. Not surprisingly, then, problematic Internet use is associated with university students' academic procrastination. Because studies are scarce in Mexico and Spain has one of the highest rates of Internet addiction in Europe, this paper (i) analyzed the presence and degree of Internet addiction among university students in Mexico and Spain, (ii) determined potential sociodemographic factors influencing Internet addiction, and (iii) established the type of correlation between Internet addiction and academic procrastination. The cross-sectional study design used an online questionnaire to measure problematic Internet use and academic procrastination through convenience sampling at one university in Mexico and one in Spain. The questionnaire contained three sections: participants' socio-demographic data, the Internet Addiction Test, and the Academic Procrastination Scale. The final sample comprised 758 university students, 387 from Mexico, and 371 from Spain, aged from 18 to 35 ($M = 20.08$, $SD = 3.16$). Results revealed similar prevalence rates of problematic and daily Internet use for leisure, potentially influencing Internet addiction in all three models (i.e., Mexico, Spain, and Total). Additionally, significant positive correlation was revealed between problematic Internet use and academic procrastination ($p < .001$). Finally, findings showed relevant data on Internet addiction's prevalence in Mexican and Spanish university contexts, along with its influential sociodemographic factors.

## Introduction

The World Wide Web, better known as the Internet, has undoubtedly contributed to society's development, facilitating communications, and becoming an essential tool in myriad jobs and professions. In recent times, however, the Internet has been massively used by the population, not only for work but also for leisure. In the last decade, leisure use has triggered remarkably

**Funding:** Financial support from WritingLab at Tecnologico de Monterrey is also gratefully acknowledged.

**Competing interests:** The authors have declared that no competing interests exist.

increased Internet addiction, influenced by social networks and affecting women more than men [1, 2]. The problem has spread worldwide, with the Internet considered the new 21st-century [3] addiction in Africa [4], Asia [5–7], North America [8], South America [9, 10], Europe [11], and Oceania [12].

Specifically, Internet addiction or problematic Internet use (PIU) affects mainly the adolescent population and university students [13–15], who are the most vulnerable to PIU, lately associated with certain risk factors. For instance, the study of Kircaburun and Griffiths [16] with university students found that being male positively correlated with participation in more gambling, more online sex, and more online betting. These risky practices were also associated with addictive behaviors that directly affected students' health [17, 18]. Other studies have shown PIU association with college students' depressive symptoms and stress [19, 20], and yet more studies have reported PIU association with young adults' alcohol and substance use [21]. Given these associations, some studies suggest that emotional regulation is a key element in assessment and treatment of Internet addiction [22].

In Mexico, the most current data indicate that, in 2019, Internet users spent 8 hours, 20 minutes on the computer daily [23]. Exceeding the 2018 figure, this shows an increasing trend in excessive Internet use. Despite the situation, few studies were conducted on Internet addiction in 2018 and 2019. Thus, the most recent study of the Mexican adolescent population showed that students do not perceive themselves as addicted to social networks [24], data that contrasts with the population's actual abusive consumption.

For their part, studies on the Mexican university environment have addressed the issue through varying approaches: (i) a study on university medical students found that Internet addiction highly correlated with somatic symptoms, anxiety, insomnia, social dysfunction, and major depression [25]; (ii) analysis of members of the National Autonomous University of Mexico found that young people have a higher rate of Internet addiction, with age an influential factor [26]; and (iii) in Tamaulipas, Mexico, approximately 9.61% of university students presented with Internet addiction [27].

This problem is accentuated in Spain because its youth population has one of the highest rates of Internet addiction in European countries [28]. Indeed, the Spanish Ministry of Health, Consumption, and Social Welfare [29] recently added "addiction to new technologies" to its Action Plan on Addictions 2018–2020, and reports indicate that 95.1% of active social network users access through their smartphones or tablets [30].

In Internet addiction among Spanish university students, the study by Fernández-Villa et al. [31] reported a PIU prevalence of 6.08% in a sample of 2,780 students. Specifically, being under 21 years old and pursuing degrees other than health sciences were influential factors for Internet addiction. However, gender was not. More recent studies have collected data alerting us to a medium-high degree of smartphone addiction among university students of education [32]. In this same population, other studies have indicated that smartphones have been the most widely used device for connecting to the network, that connecting for more than 5 hours was associated with addictive behavior, and that smartphone abuse affected men's behavior more than women's, especially in neglecting other tasks [33]. Thus, because the user spends much time surfing the Internet, accessing social networks, and watching videos on digital platforms, among other uses, neglect of tasks is consequent to PIU. Such neglect is procrastination, and in relation to academics, the term "academic procrastination" (AP) arises, meaning postponing a task until the last minute (deadline) or even being unable to complete it [34, 35].

AP is prevalent among students at all educational stages, influencing academic well-being, of course, and is linked to negative consequences including failure [36]. At the university level, AP relates to low performance and dropout [37]. Furthermore, university students are especially at risk of PIU, which reduces time spent on other activities. Several previous

investigations have reflected the link between Internet addiction and AP. In Turkish education majors, for example, significant increase was found between AP and Internet addiction [38]. In Chinese college students, Internet addiction and procrastination correlated significantly [39, 40], and, in university students in Estonia, procrastination and PIU correlated positively [41].

Having originated from these considerations, the present study was based on the theoretical model of Internet addiction [42–44], which has been extensively developed and its use consolidated and widespread, with the Internet Addiction Test as its main measurement tool [45].

Therefore, as a topic of special relevance to Internet addiction or PIU, AP is included, particularly in Mexico, because no current data exists on Internet addiction among university students, and particularly in Spain, because it is a European country with one of the highest PIU rates. Additionally, no previous studies with Spanish and Mexican university students have correlated Internet addiction with AP. Therefore, these two populations were formulated as objects of this study, to: (i) analyze the presence and degree of Internet addiction among university students in Mexico and Spain, (ii) determine potential sociodemographic factors influencing Internet addiction among university students, and (iii) establish the type of correlation between Internet addiction and AP. The following research questions were posed:

RQ1. What is the degree of Internet addiction among Mexican and Spanish university students?

RQ2. Do Mexican and Spanish university students show significant differences in Internet addiction?

RQ3. Based on sociodemographic factors, do university populations show significant differences in Internet addiction?

RQ4. Do sociodemographic factors influence Internet addiction?

RQ5. Are Internet addiction and AP statistically and significantly correlated?

## Method

### Participants and procedure

A cross-sectional study design was adopted, with a self-administered survey in a sample of undergraduate university students from the Tecnologico de Monterrey (Nuevo Leon, Mexico) ($n$ = 387) and the University of Granada (Granada, Spain) ($n$ = 371). These populations were comparable due to the students' similar socioeconomic status and the institutions' similarity in academic options.

Based on a convenience sampling design, participants' data ($N$ = 758) were collected from the questionnaire's face-to-face distribution on campus and in student e-mail lists. After receiving information about the study's purpose and anonymous data processing, participants provided informed consent and then answered questions on their sociodemographic data and on two standardized scales, one on Internet addiction and the other on AP. The data collection period was from October to December 2019.

Specifically, the Mexican sample included 178 men and 209 women, aged from 18 to 35 ($M$ = 19.59, $SD$ = 2.85); the Spanish sample included 94 men and 277 women from 18 to 35 ($M$ = 22.01, $SD$ = 3.48). Decompensation of sample of men and women in Spain is justified because the number of women enrolled in social sciences programs there is much higher than that of men [46]. Therefore, the sample size corresponds to existing reality. For age ranges, we

**Table 1. Mexican and Spanish participants' sociodemographic data.**

| | Mexico | | Spain | |
|---|---|---|---|---|
| | *n* | % | *n* | % |
| **Gender** | | | | |
| Male | 178 | 46 | 94 | 25.3 |
| Female | 209 | 54 | 277 | 74.7 |
| **Age** | | | | |
| <20 | 327 | 84.5 | 153 | 41.2 |
| 21–35 | 60 | 15.5 | 218 | 58.8 |
| **Field of knowledge** | | | | |
| Arts and Humanities | 82 | 21.2 | 48 | 12.9 |
| Sciences | 161 | 41.6 | 43 | 11.6 |
| Health Sciences | 19 | 4.9 | 48 | 12.9 |
| Social and Legal Sciences | 102 | 26.4 | 219 | 59 |
| Engineering and Architecture | 23 | 5.9 | 13 | 3.5 |
| **Marital status** | | | | |
| Single | 373 | 96.4 | 210 | 56.6 |
| Couple | 7 | 1.8 | 155 | 41.8 |
| Married | 3 | .8 | 5 | 1.3 |
| Divorced | 4 | 1 | 1 | .3 |
| **Siblings** | | | | |
| Yes | 357 | 92.2 | 337 | 90.8 |
| No | 30 | 7.8 | 34 | 9.2 |
| **Position between siblings** | | | | |
| First | 186 | 48.1 | 173 | 46.6 |
| Second | 126 | 32.6 | 144 | 38.8 |
| Third | 54 | 14 | 43 | 11.6 |
| Fourth | 15 | 3.9 | 7 | 1.9 |
| Fifth | 6 | 1.6 | 4 | 1.1 |
| **Lives with parents** | | | | |
| Yes | 283 | 73.1 | 194 | 52.3 |
| No | 104 | 26.9 | 177 | 47.7 |
| **Religious beliefs** | | | | |
| Yes | 305 | 78.8 | 141 | 38 |
| No | 82 | 21.2 | 230 | 62 |
| **Sexual orientation** | | | | |
| Heterosexual | 356 | 92 | 279 | 75.2 |
| Homosexual | 7 | 1.8 | 23 | 6.2 |
| Bisexual | 24 | 6.2 | 69 | 18.6 |
| **Number of social networks** | | | | |
| ≤2 | 11 | 2.8 | 14 | 3.8 |
| 3 | 20 | 5.2 | 29 | 7.8 |
| 4 | 23 | 5.9 | 45 | 21.1 |
| 5 | 41 | 10.6 | 40 | 10.8 |
| 6 | 65 | 16.8 | 60 | 16.2 |
| 7 | 87 | 22.5 | 69 | 18.6 |
| 8 | 68 | 17.6 | 49 | 13.2 |
| 9 | 41 | 10.6 | 36 | 9.7 |
| ≥10 | 31 | 8 | 29 | 7.8 |

(*Continued*)

**Table 1.** (Continued)

| | Mexico | | Spain | |
|---|---|---|---|---|
| | *n* | % | *n* | % |
| **Daily Internet usage time for academic purposes** | | | | |
| <1 hour | 29 | 7.5 | 40 | 10.8 |
| 1–2 hours | 95 | 24.5 | 142 | 38.3 |
| 2–3 hours | 140 | 36.2 | 118 | 31.8 |
| 3–4 hours | 79 | 20.4 | 46 | 12.4 |
| 4–5 hours | 24 | 6.2 | 14 | 3.8 |
| >5 hours | 20 | 5.2 | 11 | 3 |
| **Daily Internet usage time for leisure** | | | | |
| <1 hour | 21 | 5.4 | 20 | 5.4 |
| 1–2 hours | 66 | 17.1 | 68 | 18.3 |
| 2–3 hours | 130 | 33.6 | 127 | 34.2 |
| 3–4 hours | 98 | 25.3 | 94 | 25.3 |
| 4–5 hours | 40 | 10.3 | 34 | 9.2 |
| >5 hours | 32 | 8.3 | 28 | 7.5 |
| **Electronic device** | | | | |
| Computer | 19 | 4.9 | 13 | 3.5 |
| Laptop | 116 | 30 | 123 | 33.2 |
| Smartphone | 244 | 63 | 231 | 62.3 |
| Tablet | 8 | 2.1 | 4 | 1.1 |

chose the World Health Organization's [47] categories: ≤20 as teenager and 21–35 as young adult. Table 1 displays participants' sociodemographic data.

## Measures

**Sociodemographic measures.** Participants' sociodemographic variables included the following: country, gender, area of studies (i.e., Arts and Humanities, Sciences, Health Sciences, Social and Legal Sciences, and Engineering and Architecture), marital status, having siblings, position among siblings, living in parents' home, religious beliefs, and sexual orientation. Data were also collected on the number of social networks used, daily Internet use time for academic purposes, daily Internet use time for leisure, and type of electronic device used for daily Internet access.

**Internet Addiction Test (IAT).** Found to be a valid and reliable measure, the IAT, with 20 items, is the most commonly used instrument for measuring addiction [44, 48–50]:

1. How often do you find that you stay online longer than you intended?

2. How often do you neglect household chores to spend more time online?

3. How often do you prefer the excitement of the Internet to intimacy with your partner?

4. How often do you form new relationships with fellow online users?

5. How often do others in your life complain to you about the amount of time you spend online?

6. How often do your grades or schoolwork suffer because of the amount of time you spend online?

7. How often do you check your e-mail before something else that you need to do?

8. How often does your job performance or productivity suffer because of the Internet?

9. How often do you become defensive or secretive when anyone asks you what you do online?

10. How often do you block out disturbing thoughts about your life with soothing thoughts of the Internet?

11. How often do you find yourself anticipating when you will go online again?

12. How often do you fear that life without the Internet would be boring, empty, and joyless?

13. How often do you snap, yell, or act annoyed if someone bothers you while you are online?

14. How often do you lose sleep due to late-night logins?

15. How often do you feel preoccupied with the Internet when offline or fantasize about being online?

16. How often do you find yourself saying "just a few more minutes" when online?

17. How often do you try to cut down the amount of time you spend online and fail?

18. How often do you try to hide how long you've been online?

19. How often do you choose to spend more time online than going out with others?

20. How often do you feel depressed, moody, or nervous when you are offline, with these feelings going away once you are back online?

Based on frequency, respondents rate items on a 6-point Likert scale, with 0 = never, and 5 = always. Scale scores range from 0 to 100 points, divided by addiction ranges: 0–30 (Normal), 31–49 (Mild), 50–79 (Moderate), and 80–100 (Severe). Based on their scores, the study's participants were separated into a non-PIU group (scores < 49) and a PIU group (scores > 50) [20, 51]. In this study, the IAT scale obtained good internal consistency: Mexican sample, Cronbach's a = .884; Spanish sample, Cronbach's a = .896; Total, Cronbach's a = .889.

**Academic Procrastination Scale (APS-SV).** The Academic Procrastination Scale–Short Version (APS-SV) [52] measures academic procrastination with the following five items [53]:

1. I put off projects until the last minute.

2. I know I should work on schoolwork, but I just don't do it.

3. I get distracted by other, more fun things when I am supposed to work on schoolwork.

4. When given an assignment, I usually put it away and forget about it until it is almost due.

5. I frequently find myself putting off important deadlines.

Participants rate their agreement on a 5-point Likert scale, from 1 = disagree to 5 = agree. Scale scores range from 5 to 25 points, with higher scores indicating a greater tendency to AP. The APS-SV has good psychometric properties and internal consistency [53, 54]. For this sample, its reliability was good: Mexican sample, Cronbach's a = .885; Spanish sample, Cronbach's a = .888; Total, Cronbach's a = .888.

## Data analysis

Data were analyzed with Microsoft Excel Professional Plus 2013 (Microsoft, Redmond, WA), IBM SPSS and IBM SPSS Amos, version 24 (IBM Corp., Armonk, NY). Data were first collected in Excel, a data matrix was then created in SPSS format, and finally, data were exported to SPSS Amos.

Use of statistical tests depended on study objectives and questions. Thus, frequencies and percentages of total IAT and APS scores were established according to sociodemographic factors. Any significant differences among factors were analyzed with the t test for independent samples and the multivariate analysis of covariance (MANCOVA) test.

Additionally, linear regression analysis was performed to examine the possible influence of sociodemographic factors and AP on Internet addiction. Furthermore, prior to establishing Multi-Group Structural Equation Modeling (MG–SEM), the Mardia coefficient was calculated to confirm the hypothesis of multivariate normality of data [55]. Finally, correlation between these two variables was calculated for each population group and in total. Thus, within path analysis, Internet addiction and AP were placed as endogenous variables, and sociodemographic factors significant in any of the three models as exogenous variables.

## Results

The presence of Internet addiction in the two groups was similar, with the Mexican population revealing PIU of 11.37% and the Spanish population 12.13% (Table 2). Degrees of Internet addiction were also similar, with most of the population in the normal or mild range (88.63% in Mexico; 87.87% in Spain). However, events of severe Internet addiction appeared only in Mexico, three cases (.78%).

The t test for independent samples confirmed no statistically significant differences between IAT scores of Mexican students ($M = 32.51$, $SD = 14.81$) and Spanish students ($M = 31.05$, $SD = 15.04$) ($t = 1.34$, $df = 756$, $p = .179$). However, significant differences were found for academic procrastination: APS-SV scores for Mexican students ($M = 14.03$, $SD = 5.37$) and for Spanish students ($M = 12.41$, $SD = 5.40$) ($t = 4.12$, $df = 756$, $p = .000$).

Based on both populations' sociodemographic factors (Table 3), the greatest proportional cases were: Mexican men (7.72%); Spaniards ages 21–35 (9.36%); Spanish Engineering and Architecture (11.11%); Spanish couples (9.26%); Spanish students without siblings (12.5%); Mexican fifth children (20%); Spaniards not living with their parents (8.9%); Spaniards without religious beliefs (9.61%); Spaniards with homosexual orientation (20%); Mexicans with seven social networks (10.9%); Mexicans who dedicate from 4 to 5 hours daily to academic Internet use (13.15%); Mexicans who dedicate more than 5 hours daily to Internet leisure use (20%); and Mexicans using tablets the most to access the Internet (33.33%).

Unidirectional MANCOVA was statistically significant, with differences between countries in combined dependent variables after controlling for the Internet addiction construct (F-statistic = 53.444; $p = .000$, Wilks 'Λ = .517). This allowed further examination of group comparisons, and significant differences were found according to gender ($p = .000$), age ($p = .000$), field of knowledge ($p = .000$), marital status ($p = .000$), living with parents ($p = .000$), religious

**Table 2. Internet addiction degree in Mexican and Spanish students.**

| Internet Addiction Score | Mexico | | Spain | |
|---|---|---|---|---|
| | *n* | % | *n* | % |
| Normal range | 184 | 47.55 | 193 | 52.02 |
| Mild | 159 | 41.08 | 133 | 35.85 |
| **Total Non-PIU (< 50 scores)** | 343 | 88.63 | 326 | 87.87 |
| Moderate | 41 | 10.59 | 45 | 12.13 |
| Severe | 3 | .78 | – | – |
| **Total PIU (> 50 scores)** | 44 | 11.37 | 45 | 12.13 |

– = no event.

**Table 3. Distribution of Internet addiction cases by sociodemographic factors.**

| Variables | n (%) | Mexico | | Spain | | |
|---|---|---|---|---|---|---|
| | | NPIU (%) | PIU (%) | NPIU (%) | PIU (%) | p |
| **Gender** | | | | | | |
| Male | 272 (35.9) | 157 (57.72) | 21 (7.72) | 79 (29.04) | 15 (5.52) | .000 |
| Female | 486 (64.1) | 186 (38.27) | 23 (4.74) | 247 (50.82) | 30 (6.17) | |
| **Age** | | | | | | |
| <20 | 480 (63.3) | 289 (60.21) | 38 (7.92) | 134 (27.92) | 19 (3.95) | .000 |
| 21–35 | 278 (36.7) | 54 (19.42) | 6 (2.16) | 192 (69.06) | 26 (9.36) | |
| **Field of knowledge** | | | | | | |
| Arts and Humanities | 130 (17.2) | 72 (55.38) | 10 (7.69) | 39 (30) | 9 (6.93) | .000 |
| Sciences | 204 (26.9) | 143 (70.1) | 18 (8.82) | 36 (17.65) | 7 (3.43) | |
| Health Sciences | 67 (8.8) | 19 (28.36) | – | 45 (67.16) | 3 (4.48) | |
| Social and Legal Sciences | 321 (42.3) | 88 (27.41) | 14 (4.36) | 197 (61.38) | 22 (6.85) | |
| Engineering and Architecture | 36 (4.7) | 21 (58.33) | 2 (5.56) | 9 (25) | 4 (11.11) | |
| **Marital status** | | | | | | |
| Single | 583 (76.9) | 331 (56.77) | 42 (7.20) | 181 (31.05) | 29 (4.98) | .000 |
| Couple | 162 (21.4) | 5 (3.09) | 2 (1.23) | 140 (86.42) | 15 (9.26) | |
| Married | 8 (1.1) | 3 (37.5) | – | 4 (50) | 1 (12.5) | |
| Divorced | 5 (.7) | 4 (80) | – | 1 (20) | – | |
| **Siblings** | | | | | | |
| Yes | 694 (91.6) | 315 (45.39) | 42 (6.05) | 300 (43.23) | 37 (5.33) | .454 |
| No | 64 (8.4) | 28 (43.75) | 2 (3.13) | 26 (40.62) | 8 (12.5) | |
| **Position between siblings** | | | | | | |
| First | 359 (47.4) | 164 (45.68) | 22 (6.13) | 150 (41.78) | 23 (6.41) | .315 |
| Second | 270 (35.6) | 117 (43.33) | 9 (3.33) | 129 (47.78) | 15 (5.56) | |
| Third | 97 (12.8) | 45 (46.39) | 9 (9.28) | 38 (39.18) | 5 (5.15) | |
| Fourth | 22 (2.9) | 13 (59.09) | 2 (9.09) | 6 (27.27) | 1 (4.55) | |
| Fifth | 10 (1.3) | 4 (40) | 2 (20) | 3 (30) | 1 (10) | |
| **Lives with parents** | | | | | | |
| Yes | 477 (62.9) | 248 (52) | 35 (7.34) | 174 (36.48) | 20 (4.18) | .000 |
| No | 281 (37.1) | 95 (33.80) | 9 (3.20) | 152 (54.1) | 25 (8.9) | |
| **Religious beliefs** | | | | | | |
| Yes | 446 (58.8) | 273 (61.21) | 32 (7.17) | 126 (28.25) | 15 (3.37) | .000 |
| No | 312 (41.2) | 70 (22.44) | 12 (3.85) | 200 (64.10) | 30 (9.61) | |
| **Sexual orientation** | | | | | | |
| Heterosexual | 635 (83.8) | 318 (50.08) | 38 (5.98) | 252 (39.69) | 27 (4.25) | .000 |
| Homosexual | 30 (4) | 6 (20) | 1 (3.3) | 17 (56.7) | 6 (20) | |
| Bisexual | 93 (12.3) | 19 (20.43) | 5 (5.38) | 57 (61.29) | 12 (12.9) | |
| **Number of social networks** | | | | | | |
| ≤2 | 25 (3.3) | 11 (44) | – | 12 (48) | 2 (8) | .014 |
| 3 | 49 (6.5) | 17 (34.7) | 3 (6.12) | 29 (59.18) | – | |
| 4 | 68 (9) | 22 (32.35) | 1 (1.47) | 42 (61.76) | 3 (4.41) | |
| 5 | 81 (10.7) | 39 (48.15) | 2 (2.47) | 38 (46.91) | 2 (2.47) | |
| 6 | 125 (16.5) | 60 (48) | 5 (4) | 49 (39.2) | 11 (8.8) | |
| 7 | 156 (20.6) | 70 (44.88) | 17 (10.9) | 64 (41.02) | 5 (3.2) | |
| 8 | 117 (15.4) | 61 (52.14) | 7 (5.98) | 37 (31.62) | 12 (10.26) | |
| 9 | 77 (10.2) | 37 (48.05) | 4 (5.19) | 32 (41.57) | 4 (5.19) | |
| ≥10 | 60 (7.9) | 26 (43.34) | 5 (8.33) | 23 (38.33) | 6 (10) | |

(*Continued*)

**Table 3.** (Continued)

| Variables | n (%) | Mexico | | Spain | | |
|---|---|---|---|---|---|---|
| | | NPIU (%) | PIU (%) | NPIU (%) | PIU (%) | p |
| **Daily Internet usage time for academic purposes** | | | | | | |
| <1 hour | 69 (9.1) | 22 (31.89) | 7 (10.14) | 32 (46.38) | 8 (11.59) | .000 |
| 1–2 hours | 237 (31.3) | 87 (36.71) | 8 (3.37) | 120 (50.64) | 22 (9.28) | |
| 2–3 hours | 258 (34) | 122 (47.29) | 18 (6.98) | 109 (42.25) | 9 (3.48) | |
| 3–4 hours | 125 (16.5) | 73 (58.4) | 6 (4.8) | 44 (35.2) | 2 (1.6) | |
| 4–5 hours | 38 (5) | 19 (50) | 5 (13.15) | 10 (26.32) | 4 (10.53) | |
| >5 hours | 31 (4.1) | 20 (64.52) | – | 11 (35.48) | – | |
| **Daily Internet usage time for leisure** | | | | | | |
| <1 hour | 41 (5.4) | 21 (51.22) | – | 19 (46.34) | 1 (2.44) | .877 |
| 1–2 hours | 134 (17.7) | 63 (47.01) | 3 (2.24) | 65 (48.51) | 3 (2.24) | |
| 2–3 hours | 257 (33.9) | 114 (44.35) | 16 (6.23) | 117 (45.53) | 10 (3.89) | |
| 3–4 hours | 192 (25.3) | 91 (47.4) | 7 (3.65) | 77 (40.1) | 17 (8.85) | |
| 4–5 hours | 74 (9.8) | 34 (45.94) | 6 (8.11) | 26 (35.14) | 8 (10.81) | |
| >5 hours | 60 (7.9) | 20 (33.33) | 12 (20) | 22 (36.67) | 6 (10) | |
| **Electronic device** | | | | | | |
| Computer | 32 (4.2) | 18 (56.25) | 1 (3.13) | 9 (28.12) | 4 (12.5) | .859 |
| Laptop | 239 (31.5) | 107 (44.77) | 9 (3.77) | 111 (46.44) | 12 (5.02) | |
| Smartphone | 475 (62.7) | 214 (45.05) | 30 (6.32) | 203 (42.74) | 28 (5.89) | |
| Tablet | 12 (1.6) | 4 (33.33) | 4 (33.33) | 3 (25) | 1 (8.34) | |

p calculated through MANCOVA test;– = no event.

belief ($p = .000$), sexual orientation ($p = .000$), number of social networks ($p = .014$), and daily Internet usage time for academic purposes ($p = .000$).

The Internet addiction multiple linear regression model presented an adequate adjustment and was significant for Mexico ($R^2 = .179$; $F$-statistic = 6.270; $p = .000$), Spain ($R^2 = .204$; $F$-statistic = 7.033; $p = .000$), and Total ($R^2 = .166$; $F$-statistic = 10.599; $p = .000$) (Table 4). Significant independent variables for the Mexican model were sexual orientation ($p = .048$), leisure daily Internet ($p = .000$), and electronic device ($p = .012$); for the Spanish model: field of knowledge ($p = .007$), number of social networks ($p = .002$), academic daily Internet ($p = .041$), and daily Internet leisure ($p = .000$); for the Total model: sexual orientation ($p = .024$), number of social networks ($p = .002$), academic daily Internet ($p = .028$), and leisure daily Internet ($p = .000$).

For MG–SEM, the hypothesis of multivariate normality was fulfilled in all three models. For model 1 (Mexico), the Mardia coefficient obtained a value of 104.135, for model 2 (Spain) 89.728, and for model 3 (Total) 103.873. All were lower than p × (p + 2), where p = total number of variables (25) [56].

MG–SEM goodness-of-fit indexes were normal and confirmed the data's adequacy [57] (Table 5).

With respect to estimates, significant associations previously described in the linear regression model between independent variables and Internet addiction were established (Table 6). However, variables' influence on AP related to Internet use was also calculated. In the three models ($p = ^{***}$) and in daily Internet use for academic purposes in the Spain and Total models ($p = ^{***}$), the relationship with daily Internet use for leisure was significant. Additionally, in the three models, the correlation between Internet addiction and AP was significant ($p = ^{***}$).

**Table 4. Internet addiction multiple linear regression analysis results.**

| | Independent variable | *B* | SE | *T* | *B* | *p* |
|---|---|---|---|---|---|---|
| Mexico | Gender | −.812 | 1.492 | −.544 | −.027 | .586 |
| | Age | −.261 | 2.033 | −.128 | −.006 | .898 |
| | Field of knowledge | .007 | .590 | .013 | .001 | .990 |
| | Marital status | −3.362 | 1.963 | −1.713 | −.084 | .088 |
| | Siblings | −4.674 | 2.712 | −1.724 | −.084 | .086 |
| | Position between siblings | −.100 | .787 | −.127 | −.006 | .899 |
| | Lives with parents | −2.590 | 1.661 | −1.560 | −.078 | .120 |
| | Religious beliefs | 1.906 | 1.780 | 1.071 | .053 | .285 |
| | Sexual orientation | 2.810 | 1.417 | 1.983 | .094* | .048 |
| | Number of social networks | .376 | .374 | 1.007 | .050 | .315 |
| | Academic daily Internet | −.706 | .607 | −1.163 | −.058 | .245 |
| | Leisure daily Internet | 3.587 | .575 | 6.234 | .308*** | .000 |
| | Electronic device | 3.027 | 1.205 | 2.512 | .125* | .012 |
| Spain | Gender | .840 | 1.692 | .496 | .024 | .620 |
| | Age | −2.936 | 1.579 | −1.859 | −.096 | .064 |
| | Field of knowledge | −1.817 | .669 | −2.718 | −.137** | .007 |
| | Marital status | −1.126 | 1.346 | −.837 | −.040 | .403 |
| | Siblings | 4.836 | 2.610 | 1.837 | .093 | .065 |
| | Position between siblings | .371 | .913 | .407 | .020 | .685 |
| | Lives with parents | 2.075 | 1.496 | 1.387 | .069 | .166 |
| | Religious beliefs | .574 | 1.574 | .364 | .019 | .716 |
| | Sexual orientation | .599 | .971 | .617 | .031 | .538 |
| | Number of social networks | 1.072 | .348 | 3.078 | .154** | .002 |
| | Academic daily Internet | −1.343 | .653 | −2.056 | −.100* | .041 |
| | Leisure daily Internet | 4.004 | .618 | 6.482 | .333*** | .000 |
| | Electronic device | −1.700 | 1.313 | −1.294 | −.065 | .196 |
| Total | Country | −.367 | 1.393 | −.264 | −.012 | .792 |
| | Gender | −.254 | 1.107 | −.230 | −.008 | .818 |
| | Age | −1.199 | 1.238 | −.968 | −.039 | .333 |
| | Field of knowledge | −.715 | .435 | −1.643 | −.060 | .101 |
| | Marital status | −2.050 | 1.104 | −1.856 | −.069 | .064 |
| | Siblings | .249 | 1.880 | .133 | .005 | .895 |
| | Position between siblings | −.032 | .592 | −.054 | −.002 | .957 |
| | Lives with parents | −.233 | 1.105 | −.211 | −.008 | .833 |
| | Religious beliefs | 1.207 | 1.175 | 1.027 | .040 | .305 |
| | Sexual orientation | 1.796 | .792 | 2.269 | .081* | .024 |
| | Number of social networks | .808 | .254 | 3.186 | .112** | .002 |
| | Academic daily Internet | −.975 | .444 | −2.197 | −.077* | .028 |
| | Leisure daily Internet | 3.850 | .422 | 9.122 | .325*** | .000 |
| | Electronic device | .832 | .833 | .943 | .033 | .346 |

*p < .05
**p < .01
***p < .001.

SEM estimates for Mexico showed positive and significant correlation between Internet addiction and AP ($r = .545$; $p = ^{***}$); the coefficient of determination for Internet addiction was 15.2% ($R^2 = .152$) and for AP 6.7% ($R^2 = .067$) (Fig 1).

**Table 5. Goodness of fit measure.**

| Fit indices | Obtained values | | | Criteria |
|---|---|---|---|---|
| | **Mexico** | **Spain** | **Total** | |
| $\chi^2/df$ | 1.582 | 1.976 | .261 | $\leq 3$ |
| GFI | .998 | .992 | 1 | $\geq .90$ |
| RMSEA | .039 | .042 | .000 | $< .05$ |
| NFI | .991 | .964 | .999 | $\geq .90$ |
| CFI | .997 | .966 | 1 | $\geq .90$ |
| AGFI | .976 | .924 | .998 | $\geq .90$ |

$\chi^2$ = Chi-square; $df$ = degrees of freedom; GFI = goodness-of-fit index; RMSEA = root mean squared error of approximation; NFI = normalized fit index; CFI = comparative fit index; AGFI = adjusted goodness-of-fit index.

SEM estimates for Spain showed positive and significant correlation between Internet addiction and AP ($r = .646$; $p = ^{***}$); the coefficient of determination for Internet addiction was 17.7% ($R^2 = .177$) and for AP 13.6% ($R^2 = .136$) (Fig 2).

SEM estimates for the Total sample showed positive and significant correlation between Internet addiction and AP ($r = .597$; $p = ^{***}$); the coefficient of determination for Internet addiction was 15.3% ($R^2 = .153$) and for AP 9.2% ($R^2 = .092$) (Fig 3).

## Discussion

In both Mexico and Spain, data revealed an average Internet addiction rate of about 11.75% (RQ1). At the same time, no significant differences in the presence and degree of Internet addiction emerged between Mexican and Spanish students (RQ2). This data was relevant because Mexican students are unaware of their addictive behaviors [24] despite data similar to that of Spain, which has one of the highest rates of addiction among European countries [28]. Since 2016, when Internet addiction among Mexican university students was 9.61% and among Spanish university students 6.08%, both percentages have risen to well over 11% [27, 31]. These data warn that Internet addiction is increasing.

For AP, Mexican students had a higher average than Spanish students so were more prone to losing time. Although the populations were similar in Internet addiction, they were not comparable in AP. The differences concurred with this result in the two population groups' distinctions in subsequent statistical tests and the MANCOVA's relevance.

The ratio of cases to population size for each sociodemographic factor revealed the most cases among Mexican men, coinciding with other studies that highlight men's prevalence over women [16, 33]. In the Spanish sample, conversely, the highest rate was among women, as has been noted in other studies [1, 2]. In Spain, the age range of 21–35 was also a potential factor, suggesting the worrisome nature of university students' addiction prevalence [13–15]. Spanish engineering and architecture students showed a higher prevalence rate, previously indicated by Fernández-Villa et al. [31]. Therefore, this study fulfilled the assumption that health students have a lower rate of Internet addiction. Other potential indicators were having a partner (Spain), not having siblings (Spain), being the fifth child (Mexico), not living with parents (Spain), not having religious beliefs (Spain), being homosexual (Spain), having seven social networks (Mexico), spending 4–5 hours a day on academic Internet use (Mexico), spending more than 5 hours a day on leisure Internet use (Mexico), and the tablet as a main Internet connection device (Mexico). All these risk factors for Internet addiction increased PIU prevalence among university students in both countries.

**Table 6. Parameter estimates.**

| | Associations Between Variables | Cov | SE | CR | *p* | SRW |
|---|---|---|---|---|---|---|
| Mexico | Internet addiction ← Sexual orientation | 3.961 | 1.259 | 3.146 | .002 | .132 |
| | Internet addiction ← Leisure Internet | 3.840 | .553 | 6.943 | *** | .329 |
| | Internet addiction ← Electronic device | 2.977 | 1.148 | 2.592 | .010 | .123 |
| | AP ← Leisure Internet | 1.006 | .210 | 4.792 | *** | .238 |
| | AP ← Electronic device | .662 | .436 | 1.518 | .129 | .075 |
| | Internet addiction ↔ AP | .259 | .042 | 6.139 | *** | .545 |
| Spain | Internet addiction ← Field of knowledge | −1.778 | .627 | −2.834 | .005 | −.134 |
| | Internet addiction ← Academic Internet | −1.147 | .634 | −1.810 | .070 | −.086 |
| | Internet addiction ← Leisure Internet | 3.978 | .598 | 6.656 | *** | .331 |
| | Internet addiction ← Social networks | 1.017 | .345 | 2.944 | .003 | .146 |
| | AP ← Academic Internet | −1.053 | .233 | −4.517 | *** | −.219 |
| | AP ← Leisure Internet | 1.142 | .220 | 5.195 | *** | .264 |
| | AP ← Social networks | .00 | .127 | .786 | .432 | .040 |
| | Internet addiction ↔ AP | .325 | .047 | 6.909 | *** | .646 |
| Total | Internet addiction ← Sexual orientation | 1.927 | .661 | 2.914 | .004 | .086 |
| | Internet addiction ← Academic Internet | −.953 | .423 | −2.251 | .024 | −.075 |
| | Internet addiction ← Leisure Internet | 3.930 | .410 | 9.585 | *** | .332 |
| | Internet addiction ← Social networks | .837 | .249 | 3.362 | *** | .116 |
| | AP ← Academic Internet | −.744 | .160 | −4.659 | *** | −.162 |
| | AP ← Leisure Internet | 1.060 | .155 | 6.854 | *** | .245 |
| | AP ← Social networks | .134 | .094 | 1.431 | .152 | .051 |
| | Internet addiction ↔ AP | .305 | .033 | 9.344 | *** | .597 |

AP = academic procrastination; Cov = covariance; SE = standard error; CR = critical radio; SRW = standardized regression weights

***$p < .001$.

In students' sociodemographic characteristics, however, significant differences were found between countries in PIU (RQ3). In contrast to other studies [31], these differences occurred in gender (i), with prevalence rates higher in Mexican men and in Spanish women. As for age (ii), the most cases were ≤20 years in Mexico and 21–35 in Spain, confirming that the Mexican population tended to concentrate the most cases of Internet addiction at a young age [26]. Field of knowledge (iii) showed the most cases among Mexican science students and among Spanish social and legal science students. In marital status (iv), being single in Mexico and having a partner in Spain were indicators. In Spain, living with parents (v) seemed to increase the rate of Internet addiction, but in Mexico, the situation was reversed. Indeed, not living with parents often means the student decides what to do at each moment without imposed restrictions, possibly leading to excessive Internet use. In religious belief (vi), significant differences were found between the Mexican and Spanish populations, possibly because the Mexican population had a higher rate of believers, and the Spanish population a higher rate of non-believers. As for sexual orientation (vii), in Mexico, heterosexuals had the highest prevalence rate, but in Spain, homosexual or bisexual orientation indicated higher rates. These data are interesting for future studies, that is, to discover why this was a potentially influential factor. Obviously, a higher number of social networks (viii) generated some dependence, and PIU's prevalence was higher in students with seven or eight social networks—in fact, higher than those with 10 or more networks, probably because users with 10 or more are not as active in all their networks as those with seven or eight. The highest daily use rate in both populations was among those who spent from 4 to 5 hours on the Internet for academic purposes (ix). Finally,

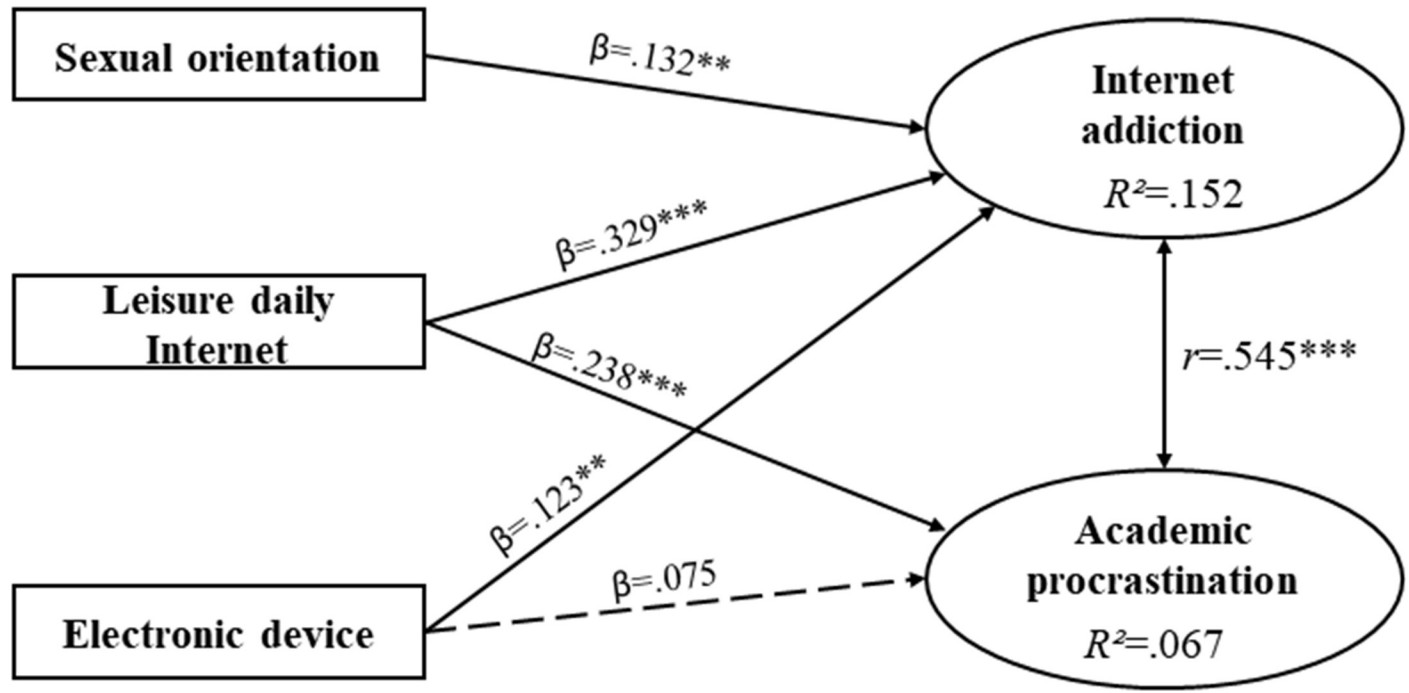

**Fig 1. Estimations of the Mexican sample's structural equation model.** β = standardized direct effect; *r* = correlation coefficient; **p < .01; ***p < .001. Discontinuous arrow = not significant.

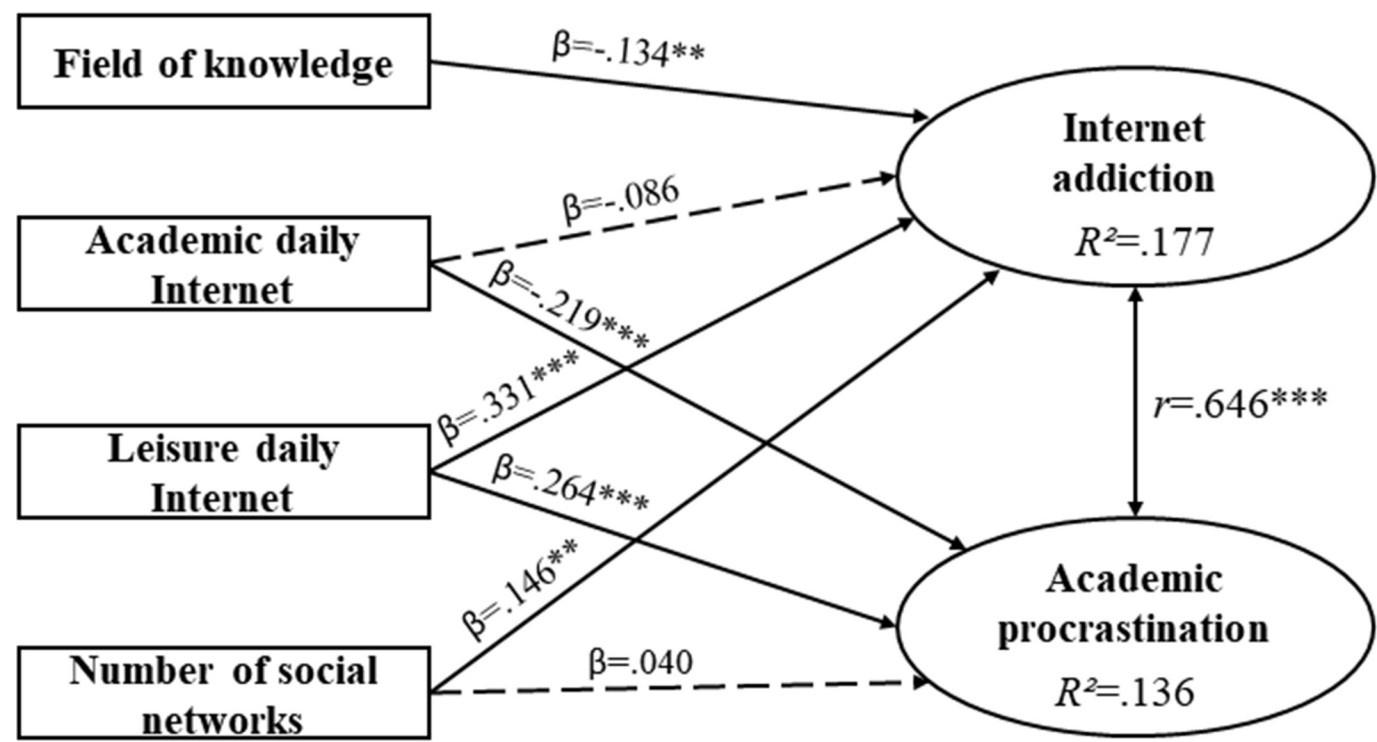

**Fig 2. Estimations of the Spanish sample's structural equation model.** β = standardized direct effect; *r* = correlation coefficient; **p < .01; ***p < .001. Discontinuous arrow = not significant.

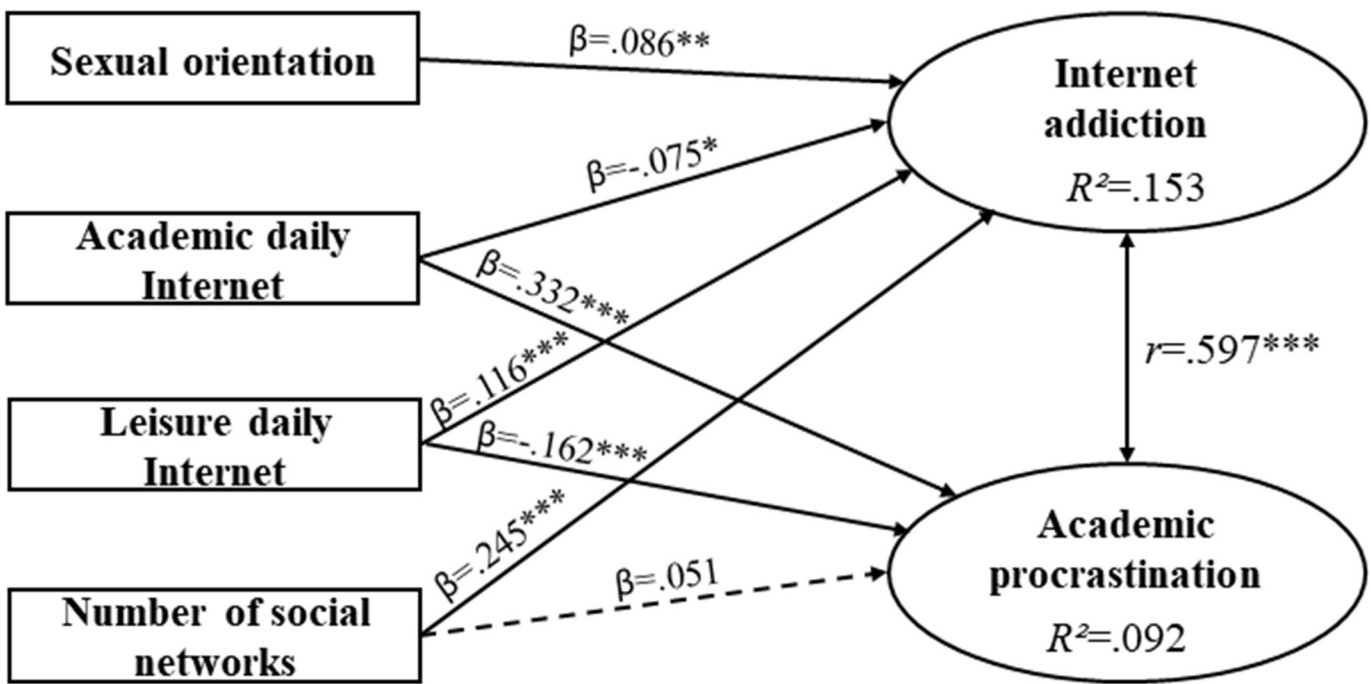

**Fig 3. Estimations of the Total sample's structural equation model.** β = standardized direct effect; r = correlation coefficient; **p < .01; ***p < .001. Discontinuous arrow = not significant.

the most students used smartphones (x) to access the Internet, not coincidentally, but because the smartphone is overall the most used device [30] and also used to access social networks [32, 33].

Among the multiple linear regression model's main findings were the following potentially influential factors for Internet addiction (RQ4): in Mexican students, sexual orientation, daily use of Internet for leisure, and the electronic device used; for Spanish students, area of knowledge, number of social networks, daily use of Internet for academic purposes, and daily use of Internet for leisure. Finally, for the population as a whole (Total model), influential factors were sexual orientation, number of social networks, daily use of Internet for academic purposes, and daily use of Internet for leisure. The three models' only coinciding factor was daily use of Internet for leisure, with a prevalence indicator of more than 5 hours a day, following Ruiz-Palmero et al. [33]. Other factors were unique to each study model. Although, due to their cross-sectional nature, these indicators are not conclusive data in the Internet addiction construct, they are potentially influential factors for Mexican and Spanish students.

In all three models, significant and positive correlations were established between Internet addiction and AP (RQ5). Thus, the greater the Internet addiction, the greater the procrastination, and vice versa. Therefore, these data confirmed study findings from Turkey, Estonia, and China [38–41], thus broadening knowledge of this problem in the Mexican and Spanish contexts, under the theoretical framework of Internet addiction [42–44].

## Limitations and implications

The study's cross-sectional nature and convenience sampling are highlighted as limitations. Because it is a transversal study, a causal link between Internet addiction and AP cannot be inferred. This inference of the influence of constructs responds to a specific moment. Therefore, such casuistry can be tested if repeated over time in future longitudinal studies.

Furthermore, because this study was conducted at two specific universities, generalization of the results is limited, and future studies should collect data from various universities in the two countries.

## Conclusions

Internet addiction is a current global problem. Specifically, studies focusing on the Mexican context are scarce, and more research is needed in Spain where PIU of is of the highest risk. This research has addressed various objectives to advance knowledge about the problem's presence and degree in two populations varying geographically, but similar in data. The study has identified various sociodemographic factors as potential indicators of Internet addiction. At the same time, information has been collected on the correlation between Internet addiction and AP in Mexican and Spanish university students. Additionally, the purposes' achievement was addressed through answers to each research question in the discussion.

All this leads us to rethink future lines of research in which the focus continues to grow and the study sample to increase, while we count on other countries and compare results among them. Therefore, we encourage studies that continue this line and replicate results in other contexts to generate strong networks and shared data on Internet addiction in university students and also in underage populations. Finally, much research remains to be done because Internet addiction, already classified as a disease, especially affects young populations, so investigating possible causes to establish preventive measures is crucial.

## Supporting information

**S1 Dataset.**
(SAV)

## Acknowledgments

This research was conducted within the framework of the pre-doctoral mobility link between the Doctorate Program in Educational Innovation at the Tecnologico de Monterrey and the Doctorate Program in Educational Sciences at the University of Granada (Reference: EST18/00046).

## Author Contributions

**Conceptualization:** Inmaculada Aznar-Díaz, José-María Romero-Rodríguez, Abel García-González, María-Soledad Ramírez-Montoya.

**Data curation:** Abel García-González, María-Soledad Ramírez-Montoya.

**Formal analysis:** Inmaculada Aznar-Díaz, José-María Romero-Rodríguez, Abel García-González.

**Investigation:** Inmaculada Aznar-Díaz, José-María Romero-Rodríguez, Abel García-González, María-Soledad Ramírez-Montoya.

**Methodology:** Inmaculada Aznar-Díaz, José-María Romero-Rodríguez, Abel García-González, María-Soledad Ramírez-Montoya.

**Software:** José-María Romero-Rodríguez.

**Supervision:** Inmaculada Aznar-Díaz, José-María Romero-Rodríguez, Abel García-González, María-Soledad Ramírez-Montoya.

**Validation:** Inmaculada Aznar-Díaz, María-Soledad Ramírez-Montoya.

**Visualization:** Abel García-González.

**Writing – original draft:** Inmaculada Aznar-Díaz, José-María Romero-Rodríguez, Abel García-González, María-Soledad Ramírez-Montoya.

**Writing – review & editing:** Inmaculada Aznar-Díaz, José-María Romero-Rodríguez, Abel García-González, María-Soledad Ramírez-Montoya.

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
