## [Decision Letter · Decision Letter 0]

31 Mar 2020

PONE-D-19-35420

Internet addiction and academic procrastination in Mexican and Spanish university students. Correlation and Predictive Factors

PLOS ONE

Dear Authors,

Thank you for submitting your manuscript to PLOS ONE. After careful consideration, we feel that it has merit but does not fully meet PLOS ONE’s publication criteria as it currently stands. Therefore, we invite you to submit a revised version of the manuscript that addresses the points raised during the review process.

We would appreciate receiving your revised manuscript by 4/29/20. To enhance the reproducibility of your results, we recommend that if applicable you deposit your laboratory protocols in protocols.io, where a protocol can be assigned its own identifier (DOI) such that it can be cited independently in the future. For instructions see: http://journals.plos.org/plosone/s/submission-guidelines#loc-laboratory-protocols

We look forward to receiving your revised manuscript.

Kind regards,

Luca Cerniglia, PhD

Academic Editor

PLOS ONE

Journal Requirements:

Reviewers' comments:

Reviewer's Responses to Questions

**Comments to the Author**

1. Is the manuscript technically sound, and do the data support the conclusions?

Reviewer #1: Partly

Reviewer #2: Partly

2. Has the statistical analysis been performed appropriately and rigorously? 

Reviewer #1: No

Reviewer #2: Yes

3. Have the authors made all data underlying the findings in their manuscript fully available?

Reviewer #1: No

Reviewer #2: Yes

4. Is the manuscript presented in an intelligible fashion and written in standard English?

Reviewer #1: No

Reviewer #2: No

5. Review Comments to the Author

Reviewer #1: see the attachment.

Reviewer #2: Thank you very much for the possibility to review the manuscript titled “Internet addiction and

academic procrastination in Mexican and Spanish university students. Correlation and Predictive

Factors”. This cross-sectional study evaluated the presence and degree of Internet addiction among university students in Mexico and Spain, and the socio-demographic factors that influence Internet addiction, establishing the kind of correlation generated between Internet addiction and academic procrastination.

I think this study is very interesting and should be published if the authors would like to make some revisions. Furthermore the paper has many grammatical errors and uncommon phrases and the manuscript should be edited by a professional native speaker.

Abstract

Authors are invited to delete the acronyms “IAT” and “APS-SV” from the abstract, as they are not necessary in the text.

Introduction

The authors have included several studies of recent literature but it is not clear which theoretical model is underlying it. In fact, it would be important to be able to make more explicit a theoretical model. Consequently, authors are invited to formulate specific hypotheses, based on the literature, instead of research questions.

Furthermore, in the introduction, the authors focused on adolescents and university students. Authors are invited to be more focused and more consistent.

In this regard, authors are advised to explore some of the relevant studies more closely:

- Thomas, M., & Tripathi, P. (2019). Comparison of internet addiction between teenagers and young adults. Indian Journal of Health & Wellbeing, 10.

- Cerniglia et al. (2019). A latent profile approach for the study of internet gaming disorder, social media addiction, and psychopathology in a normative sample of adolescents. Psychology research and behavior management, 12, 651.

- Ballarotto et al. (2018). Adolescent Internet abuse: A study on the role of attachment to parents and peers in a large community sample. BioMed research international, 2018.

- Lyvers, et al. (2016). Traits associated with internet addiction in young adults: Potential risk factors. Addictive behaviors reports, 3, 56-60.

- Cimino et al. 2018. A longitudinal study for the empirical validation of an etiopathogenetic model of internet addiction in adolescence based on early emotion regulation

Method

Sample was composed by university students. Are they all ungraduated students? Or are they also doctoral students, post graduate etc?

Participants over 36 years of age are very few and could be eliminated from the sample in order to make it more homogeneous.

As the present study is a cross-sectional study, terms indicating a causal effect should be avoided, as these are not studies that can verify these effects (e.g. longitudinal studies).

Authors are invited to include examples of items from the different tools

Discussion

As highlighted for the introductory section, it would be important to focus the discussions more closely. The results should be discussed more closely, also referring to a basic theoretical model.

6. PLOS authors have the option to publish the peer review history of their article (what does this mean?). If published, this will include your full peer review and any attached files.

Reviewer #1: No

Reviewer #2: No

---

## [Author Response · Author response to Decision Letter 0]

30 Apr 2020

Reviewer 1

Point 1: Considering the cross-sectional nature of this study, please omit “predict” and “predictive” throughout the manuscript. This is because such a design cannot infer the causal linkage under investigation. Please also elaborate more on this issue in the Limitations (although the authors only refer to the correlation in SEM model).

Response 1: The concept predict or predictive has been replaced by potential influence. This has also been added in the limitations:

In this sense, as it is a transversal study, the causal link between Internet addiction and academic procrastination cannot be inferred. This inference about the influence of constructs responds to a specific moment. Therefore, in future longitudinal studies this casuistry can be tested if it is repeated over time.

Point 2: The manuscript should be edited by a professional native speaker, as many grammatical errors and uncommon phrases have been identified throughout the paper.

Response 2: The manuscript has been revised by the team of translators at Tecnológico de Monterrey, so any previous grammatical errors have been corrected.

Point 3: What does it mean by university students exactly? Were all undergraduate, postgraduate, or doctoral students considered? Why is the sampling so broad? Concerning the means and standardized deviations of the table, it indicates that participants across two countries aged 47 and 58 years are considered as possible outliers.

Response 3: It has been specified in the sample that these are undergraduate university students. As a recommendation of ambitious reviewers, participants older than 36 years have been eliminated, as the sample size was very small and generated outliers. So the sample has remained homogeneous.

Point 4: 13. In the introduction, the authors always shift among high school students, adolescents, and university students? Since high school students and adolescents are not the focus of this study, please be selective and concentrate when conducting the literature review.

Response 4: All information relating to secondary school students and adolescents has been deleted, focusing the review on the study population (university students).

Point 5: 12. Introduction, the authors mentioned, “this problem has spread globally in developed countries” This is inaccurate, as the similar patterns have also been documented in many developing countries

Response 5: It has been deleted in developed countries so as not to limit the statement to only these types of countries.

Point 6: Are these two selected universities comparative in terms of socioeconomic status and study background? It may be the case that, before this investigation, they have already shown some inherent differences.

Response 6: Yes, they are comparable in terms of socioeconomic status and type of students. This has been added in the sample section:

These populations were comparable due to the similar socioeconomic status of the students and the similarity of the institutions with respect to the diversity of academic options they possess.

Point 7: How do the authors ensure the cross-cultural equivalence of the measurement without running multi-group CFA? Relatedly, I was wondering why the authors do not conduct MG-SEM to investigate any association differences between the two countries.

Response 7: An MG-SEM of three models has been included: Mexican, Spanish and total population (since no significant differences were found between both populations in the scale of Internet addiction).

Point 8: Based on the table 1, gender is not fully balanced, particularly in the subsamples of participants from Spain. Why? Moreover, several sociodemographic characteristic are not fully balanced; instead, the authors do not consider including these as confounding variables.

Response 8: In Spain the population of women is much larger than that of men in university degrees in social sciences. This has been specified in the sample and supported by citations. The cases older than 36 years have been eliminated since they presented a scarce number of subjects. The cases of 1 social network and 2 social networks have been grouped together in ≤2, and eliminated some sociodemographic factors with low cases such as having children and the use of social networks.

Point 9: Please add the item examples for each questionnaire (PP. 10-11).

Response 9: Items from both scales have been included.

Point 10: Please elaborate more on the rationale of selecting these model fit indices (P. 12).

Response 10: Goodness-of-fit indices have been justified and the most usual ones have been used for path analysis studies: X2/df, RMSEA, GFI, NFI, CFI and AGFI.

Point 11: What does it mean by gl exactly? (P. 13)

Response 11: gl are the degrees of freedom. This acronym was not translated into English. It has already been put as df.

Point 12: What is RMR? May you indicate SRMR? (P. 15) 

Response 12: The goodness-of-fit indexes have been re-established.

Point 13: Is any missing data involved in the present research? How do the authors handle them in the further course of data analysis?

Response 13: There are no missing data, absolutely all of them have been added. With the restructuring that has taken place, the quality of the manuscript has increased considerably.

Point 14: When comparing the internet addiction between two countries, the authors fail to consider the sociodemographic variables that may potentially influence the mean level differences. In a sense, MANCOVA should be administrated (P. 13).

Response 14: The MANCOVA has been used to compare these differences.

Point 15: The fit indices of SEM do not show that the model fits the data well (X2/df = 5.63, and pvalue is significant; P. 16). This is a significant concern.

Response 15: The value has changed when modifying the data and establishing the three models of the MG-SEM. The settings obtained in all models have been adequate.

Point 16: Overall, the discussion is poorly addressed by only two pages. I highly encourage the authors to discuss thoughtfully and more in depth concerning each purpose of this study.

Response 16: The discussion has been extended with the new results and has been approached with a greater degree of depth, referring to each objective and RQ.

Point 17: Please add the new section of Limitations and Implications, and remove the limitations from the conclusion section. Limitations are not conclusions; rather, they should be addressed in the discussion section.

Response 17: The limitations have been removed from the section on conclusions and moved to the discussion section.

Point 18: The figure provided is unclear. Moreover, according to this figure, some factor loadings are inappropriate. Why are they still being considered in further analysis?

Response 18: The figure has been modified due to the change in the sample size and the performance of a MG-SEM. Therefore, these values are no longer a problem. A path analysis has been carried out due to the relevance for the study.

 

Reviewer 2

Point 1: Authors are invited to delete the acronyms “IAT” and “APS-SV” from the abstract, as they are not necessary in the text.

Response 1: They have been removed from the abstract.

Point 2: The authors have included several studies of recent literature but it is not clear which theoretical model is underlying it. In fact, it would be important to be able to make more explicit a theoretical model. Consequently, authors are invited to formulate specific hypotheses, based on the literature, instead of research questions.

Response 2: Given the nature of the study and the research tradition of the educational sciences, it has been decided to keep the research questions rather than formulate hypotheses. With respect to the theoretical model, the theoretical model of Internet addiction on which the study is based has been explicitly added:

Based on these considerations, the present study was based on the theoretical model of Internet addiction (Goldberg, 1995; Kandell, 1998; Young, 1998). This model has been further developed in the scientific field and its use is the most widespread and consolidated, where the Internet Addiction Test is used as the main instrument (Aznar et al., 2020).

Point 3: Furthermore, in the introduction, the authors focused on adolescents and university students. Authors are invited to be more focused and more consistent. In this regard, authors are advised to explore some of the relevant studies more closely:

- Thomas, M., & Tripathi, P. (2019). Comparison of internet addiction between teenagers and young adults. Indian Journal of Health & Wellbeing, 10.

- Cerniglia et al. (2019). A latent profile approach for the study of internet gaming disorder, social media addiction, and psychopathology in a normative sample of adolescents. Psychology research and behavior management, 12, 651.

- Ballarotto et al. (2018). Adolescent Internet abuse: A study on the role of attachment to parents and peers in a large community sample. BioMed research international, 2018.

- Lyvers, et al. (2016). Traits associated with internet addiction in young adults: Potential risk factors. Addictive behaviors reports, 3, 56-60.

- Cimino et al. 2018. A longitudinal study for the empirical validation of an etiopathogenetic model of internet addiction in adolescence based on early emotion regulation

Response 3: All information relating to secondary school students and adolescents has been deleted, focusing the review on the study population (university students). Furthermore, all suggested references have been reviewed and included.

Point 4: Sample was composed by university students. Are they all ungraduated students? Or are they also doctoral students, post graduate etc?

Response 4: It has been specified in the sample that these are undergraduate university students.

Point 5: Participants over 36 years of age are very few and could be eliminated from the sample in order to make it more homogeneous.

Response 5: As a recommendation of ambitious reviewers, participants older than 36 years have been eliminated, as the sample size was very small and generated outliers. So the sample has remained homogeneous.

Point 6: As the present study is a cross-sectional study, terms indicating a causal effect should be avoided, as these are not studies that can verify these effects (e.g. longitudinal studies).

Response 6: Concepts that indicated causality (such as the concept of predicting) have been modified, by potential influence. Emphasizing the limitations:

In this sense, as it is a transversal study, the causal link between Internet addiction and academic procrastination cannot be inferred. This inference about the influence of constructs responds to a specific moment. Therefore, in future longitudinal studies this casuistry can be tested if it is repeated over time.

Point 7: Authors are invited to include examples of items from the different tools.

Response 7: Items from both scales have been included.

Point 8: As highlighted for the introductory section, it would be important to focus the discussions more closely. The results should be discussed more closely, also referring to a basic theoretical model.

Response 8: The discussion has been extended with the new results and has been approached with a greater degree of depth, referring to each objective and RQ.

---

## [Editor Report · Decision Letter 1]

11 May 2020

Mexican and Spanish university students’ Internet addiction and academic procrastination: Correlation and potential factors

PONE-D-19-35420R1

Dear Authors,

We are pleased to inform you that your manuscript has been judged scientifically suitable for publication and will be formally accepted for publication once it complies with all outstanding technical requirements.

With kind regards,

Luca Cerniglia, PhD

Academic Editor

PLOS ONE

Additional Editor Comments (optional):

The authors have addressed all the points suggested by the reviewers. I think the manuscript can be published in the present form.
---

## [Editor Report · Acceptance letter]

13 May 2020

PONE-D-19-35420R1 

Mexican and Spanish university students’ Internet addiction and academic procrastination: Correlation and potential factors 

Dear Dr. Romero-Rodríguez:

I am pleased to inform you that your manuscript has been deemed suitable for publication in PLOS ONE. Congratulations! Your manuscript is now with our production department. 

With kind regards,

on behalf of

Dr. Luca Cerniglia 

Academic Editor

PLOS ONE